# Targeted sampling of enlarged neighborhood via Monte Carlo tree search for TSP

## Abstract

The travelling salesman problem (TSP) is a well-known combinatorial optimization problem with a variety of real-life applications. We tackle TSP by incorporating machine learning methodology and leveraging the variable neighborhood search strategy. More precisely, the search process is considered as a Markov decision process (MDP), where a 2-opt local search is used to search within a small neighborhood, while a Monte Carlo tree search (MCTS) method (which iterates through simulation, selection and back-propagation steps), is used to sample a number of targeted actions within an enlarged neighborhood. This new paradigm clearly distinguishes itself from the existing machine learning (ML) based paradigms for solving the TSP, which either uses an end-to-end ML model, or simply applies traditional techniques after ML for post optimization. Experiments based on two public data sets show that, our approach clearly dominates all the existing learning based TSP algorithms in terms of performance, demonstrating its high potential on the TSP. More importantly, as a general framework without complicated hand-crafted rules, it can be readily extended to many other combinatorial optimization problems.

## 1 Introduction

The travelling salesman problem (TSP) is a well-known combinatorial optimization problem with various real-life applications, such as transportation, logistics, biology, circuit design. Given $n$ cities as well as the distance $d_{ij}$ between each pair of cities $i$ and $j$, the TSP aims to find a cheapest tour which starts from a beginning city (arbitrarily chosen), visits each city exactly once, and finally returns to the beginning city. This problem is NP-hard, thus being extremely difficult from the viewpoint of theoretical computer science.

Due to its importance in both theory and practice, many algorithms have been developed for the TSP, mostly based on traditional operations research (OR) methods. Among the existing TSP algorithms, the best exact solver Concorde (Applegate et al., 2009) succeeded in demonstrating optimality of an Euclidean TSP instance with 85,900 cities, while the leading heuristics (Helsgaun, 2017) and (Taillard & Helsgaun, 2019) are capable of obtaining near-optimal solutions for instances with millions of cities. However, these algorithms are very complicated, which generally consist of many hand-crafted rules and heavily rely on expert knowledge, thus being difficult to generalize to other combinatorial optimization problems.

To overcome those limitations, recent years have seen a number of ML based algorithms being proposed for the TSP (briefly reviewed in the next section), which attempt to automate the search process by learning mechanisms. This type of methods do not rely on expert knowledge, can be easily generalized to various combinatorial optimization problems, thus become promising research direction at the intersection of ML and OR. For the TSP, existing ML based algorithms can be roughly classified into two paradigms, i.e.: (1) **End-to-end ML paradigm** which uses a ML model alone to directly convert the input instance to a solution. (2) **ML followed by OR paradigm** which applies ML at first to provide some additional information, to guide the following OR procedure towards promising regions.

Despite its high potential, for the TSP, existing ML based methods are still in its infancy, struggling to solve instances with more than 100 cities, leaving much room for further improvement compared with traditional methods. To this end, we propose a novel framework by combining Monte Carlo

tree search (MCTS) with a basic OR method (2-opt based local search) using variable neighborhood strategy to solve the TSP. The main contributions are summarized as follows.

- **Framework**: We propose a new paradigm which combines OR and ML using variable neighborhood strategy. Starting from an initial state, a basic 2-opt based local search is firstly used to search within a small neighborhood. When no improvement is possible within the small neighborhood, the search turns into an enlarged neighborhood, where a reinforcement learning (RL) based method is used to identify a sample of promising actions, and iteratively choose one action to apply. Under this new paradigm, OR and ML respectively work within disjoint space, being flexible and targeted, and clearly different from the two paradigms mentioned above. More importantly, as a general framework without complicated hand-crafted rules, this framework could not only be applied to the TSP, but also be easily extended to many other combinatorial optimization problems.

- **Methodology**: When we search within an enlarged neighborhood, it is infeasible to enumerate all the actions. We then seek to sample a number of promising actions. To do this automatically, we implement a MCTS method which iterates through simulation, selection and back-propagation steps, to collect useful information that guides the sampling process.

  To the best of our knowledge, there is only one existing paper (Shimomura & Takashima, 2016) which also uses MCTS to solve the TSP. However, their method is a constructive approach, where each state is a partial TSP tour, and each action adds a city to increase the partial tour, until forming a complete tour. By contrast, our MCTS method is a conversion based approach, where each state is a complete TSP tour, and each action converts the original state to a new state (also a complete TSP tour). The methodology and implementation details of our MCTS are very different from the MCTS method developed in (Shimomura & Takashima, 2016).

- **Results**: We carry out experiments on two sets of public TSP instances. Experimental results (detailed in Section 4) show that, on both data sets our MCTS algorithm obtains (within reasonable time) statistically much better results with respect to all the existing learning based algorithms. These results clearly indicate the potential of our new method for solving the TSP.

The rest of this paper is organized as follows: Section 2 briefly reviews the existing learning based methods on the TSP. Section 3 describes in detail the new paradigm and the MCTS method. Section 4 provides and analyzes the experimental results. Finally Section 5 concludes this paper.

## 2   RELATED WORKS

In this section, we briefly review the existing ML based algorithms on the TSP, and then extend to several other highly related problems. Non-learned methods are omitted, and interested readers please find in (Applegate et al., 2009), (Rego et al., 2011), (Helsgaun, 2017) and (Taillard & Helsgaun, 2019) for an overlook of the leading TSP algorithms.

The idea of applying ML to solve the TSP is not new, dated back to several decades ago. Hopfield & Tank (1985) proposed a Hopfield-network, which achieved the best TSP solutions at that time. Encouraged by this progress, neural networks were subsequently applied on many related problems (surveyed by Smith (1999)). However, these early attempts only achieved limited success, with respect to other state-of-the-art algorithms, possibly due to the lack of high-performance hardware and training data. In recent years, benefited from the rapidly improving hardware and exponentially increasing data, ML (especially deep learning) achieved great successes in the field of artificial intelligence. Motivated by these successes, ML becomes again a hot and promising topic for combinatorial optimization, especially for the TSP. A number of ML based algorithms have been developed for the TSP, which can be roughly classified into two paradigms (possibly with overlaps).

**End-to-end ML paradigm:** Vinyals et al. (2015) introduced a pointer network which consists of an encoder and a decoder, both using recurrent neural network (RNN). The encoder parses each TSP city into an embedding, and then the decoder uses an attention model to predict the probability distribution over the candidate (unvisited) cities. This process is repeated to choose a city one by one, until forming a complete TSP tour. The biggest advantage of the pointer network is its ability of

processing graphs with different sizes. However, as a supervised learning (SL) method, it requires a large number of pre-computed optimal (at least high-quality) TSP solutions, being unaffordable for large-scale instances. To overcome this drawback, several successors chose reinforcement learning (RL) instead of SL, thus avoiding the requirement of pre-computed solutions. For example, Bello et al. (2017) implemented an actor-critic RL architecture, which uses the solution quality (tour length) as a reward signal, to guide the search towards promising area. Khalil et al. (2017) proposed a framework which maintains a partial tour and repeatedly calls a RL based model to select the most relevant city to add to the partial tour, until forming a complete tour. Emami & Ranka (2018) also implemented an actor-critic neural network, and chose Sinkhorn policy gradient to learn policies by approximating a double stochastic matrix. Concurrently, Deudon et al. (2018) and Kool et al. (2019) both proposed a graph attention network (GAN), which incorporates attention mechanism with RL to auto-regressively improve the quality of the obtained solution.

Specifically, Shimomura & Takashima (2016) proposed a MCTS algorithm for the TSP, which also belongs to this paradigm. As explained in the introduction section, this existing method is clearly different from our MCTS algorithm (detailed in Section 3.5).

**ML followed by OR paradigm:** Applying ML alone is difficult to achieve satisfactory performance, thus it is recommended to combine ML and OR to form hybrid algorithms (Bengio et al., 2018). Following this idea, Nowak et al. (2017) proposed a supervised approach, which trains a graph neural network (GNN) to predict an adjacency matrix (heat map) over the cities, and then attempts to convert the adjacency matrix to a feasible TSP tour by beam search (OR based method). Joshi et al. (2019) followed this framework, but chose deep graph convolutional networks (GCN) instead of GNN to build heat map, and then tried to construct tours via highly parallelized beam search. Additionally, several algorithms belonging to above paradigm were further enhanced with OR based algorithms, thus also belonging to this paradigm. For example, the solution obtained by ML is post-optimized by sampling in (Bello et al., 2017) and (Kool et al., 2019), or by 2-opt based local search in (Deudon et al., 2018). Overall, this hybrid paradigm performs statistically much better than the end-to-end ML paradigm, showing the advantage of combining ML with OR.

In addition to the works focused on the classic TSP, there are several ML based methods recently proposed for solving other related problems, such as the decision TSP (Prates et al., 2019), the multiple TSP (Kaempfer & Wolf, 2019), and the vehicle routing problem (Nazari et al., 2018), etc.

Finally, for an overall survey about machine learning for combinatorial optimization, please refer to (Bengio et al., 2018) and (Guo et al., 2019).

## 3 METHOD

### 3.1 FRAMEWORK

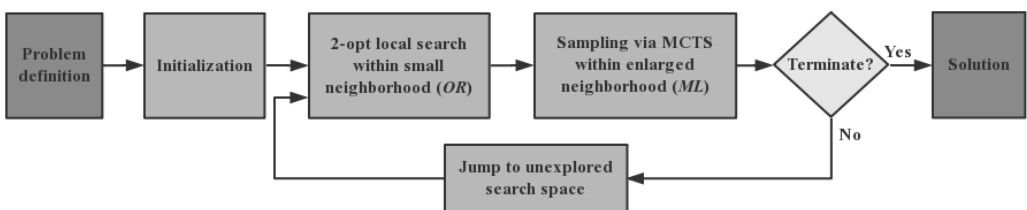

Figure 1: New paradigm for combining OR and ML to solve the TSP

The proposed new paradigm for combining OR and ML to solve the TSP is outlined in Fig. 1. Specifically, the search process is considered as a Markov Decision Process (MDP), which starts from an initial state $\pi$, and iteratively applies an action $a$ to reach a new state $\pi^*$. At first, the MDP explores within a small neighborhood, and tries to improve the current state by applying 2-opt based local search. When no further improvement is possible within the small neighborhood, the MDP turns into an enlarged neighborhood, which consists of a large number of possible actions, being infeasible to enumerate one by one. To improve search efficiency, MCTS is launched to iteratively sample a number of promising actions and choose an improving action to apply. When MCTS fails

to find an improving action, the MDP jumps into an unexplored search space, and launches a new round of 2-opt local search and MCTS again. This process is repeated until the termination condition is met, then the best found state is returned as the final solution.

## 3.2 STATES AND ACTIONS

In our implementation, each state corresponds to a complete TSP solution, i.e., a permutation $\pi = (\pi_1, \pi_2, \ldots, \pi_n)$ of all the cities. Each action $a$ is a transformation which converts a given state $\pi$ to a new state $\pi^*$. Since each TSP solution consists of a subset of $n$ edges, each action could be viewed as a $k$-opt ($2 \leq k \leq n$) transformation, which deletes $k$ edges at first, and then adds $k$ different edges to form a new tour.

Obviously, each action can be represented as a series of $2k$ sub-decisions ($k$ edges to delete and $k$ edges to add). This representation method is straightforward, but seems a bit redundant, since the deleted edges and added edges are highly relevant, while arbitrarily deleting $k$ edges and adding $k$ edges may result in an unfeasible solution. To overcome this drawback, we develop a compact method to represent an action, which consists of only $k$ sub-decisions, as exemplified in Fig. 2.

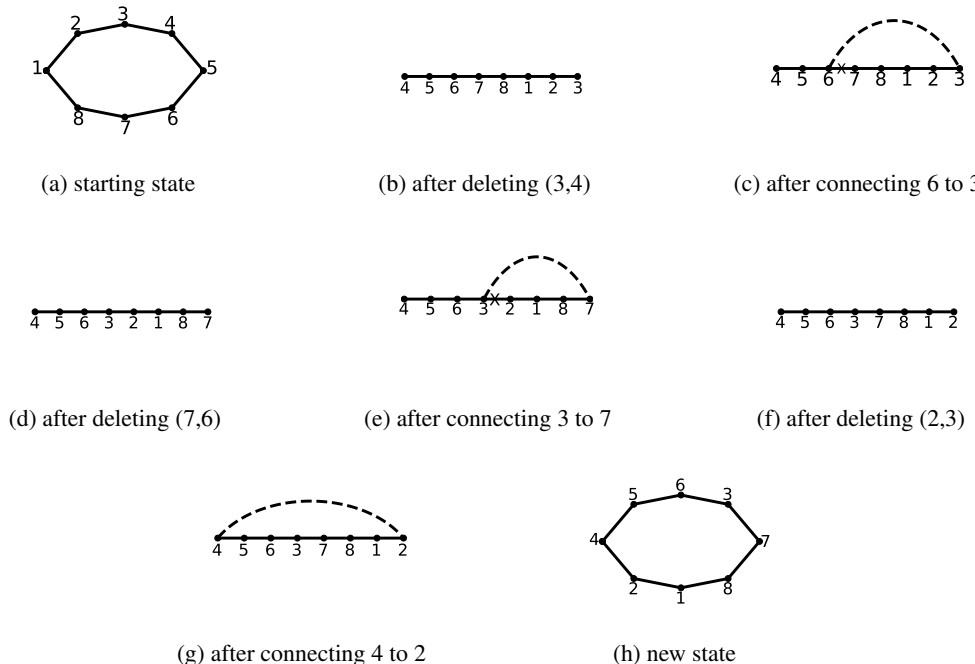

Figure 2: The decision process of an example action

In Fig. 2, sub-figure (a) is the starting state $\pi = (1, 2, 3, 4, 5, 6, 7, 8)$. To determine an action, we at first decide a city $a_1$ and delete the edge between $a_1$ and its previous city $b_1$. Without loss of generality, suppose $a_1 = 4$, then $b_1 = 3$ and edge $(3, 4)$ is deleted, resulting in a temporary status shown in sub-figure (b) (for the sake of clarity, drawn as a line which starts from $a_1$ and ends at $b_1$). Furthermore, we decide a city $a_2$ to connect city $b_1$, generally resulting in an unfeasible solution containing an inner cycle (unless $a_2 = a_1$). For example, suppose $a_2 = 6$ and connect it to city 3, the resulting temporary status is shown in sub-figure (c), where an inner cycle occurs and the degree of city $a_2$ increases to 3. To break inner cycle and reduce the degree of $a_2$ to 2, the edge between city $a_2$ and city $b_2 = 7$ should be deleted, resulting in a temporary status shown in sub-figure (d). This process is repeated, to get a series of cities $a_k$ and $b_k$ ($k \geq 2$). In this example, $a_3 = 3$ and $b_3 = 2$, respectively corresponding to sub-figures (e) and (f). Once $a_k = a_1$, the loop closes and reaches a new state (feasible TSP solution). For example, if let $a_4 = a_1 = 4$ and connect $a_4$ to $b_3$, the resulting new state is shown in sub-figure (g), which is redrawn as a cycle in sub-figure (h).

Formally, an action can be represented as $\boldsymbol{a} = (a_1, b_1, a_2, b_2, \ldots, a_k, b_k, a_{k+1})$, where $k$ is a variable and the begin city must coincide with the final city, i.e. $a_{k+1} = a_1$. Each action corresponds to a $k$-opt transformation, which deletes $k$ edges, i.e., $(a_i, b_i), 1 \leq i \leq k$, and adds $k$ edges, i.e., $(b_i, a_{i+1}), 1 \leq i \leq k$, to reach a new state. Notice that not all these elements are optional, since once $a_i$ is known, $b_i$ can be uniquely determined without any optional choice. Therefore, to determine an action we should only decide a series of $k$ sub-decisions, i.e., the $k$ cities $a_i, 1 \leq i \leq k$. Intuitively, this compact representation method brings advantages in two-folds: (1) fewer (only $k$, not $2k$) sub-decisions need to be made; (2) the resulting states are necessarily feasible solutions.

Furthermore, if $a_{i+1}$ does not belong to the top 10 nearest neighbors of $b_i$, edge $(b_i, a_{i+1})$ is marked as an unpromising edge, and any action involving $(b_i, a_{i+1})$ is marked as an unpromising action. All the unpromising actions are eliminated directly, to reduce the scale of search space.

Let $L(\boldsymbol{\pi})$ denote the tour length corresponding to state $\boldsymbol{\pi}$, then corresponding to each action $\boldsymbol{a} = (a_1, b_1, a_2, b_2, \ldots, a_k, b_k, a_{k+1})$ which converts $\boldsymbol{\pi}$ to a new state $\boldsymbol{\pi}^*$, the difference $\Delta(\boldsymbol{\pi}, \boldsymbol{\pi}^*) = L(\boldsymbol{\pi}^*) - L(\boldsymbol{\pi})$ could be calculated as follows:

$$\Delta(\boldsymbol{\pi}, \boldsymbol{\pi}^*) = \sum_{i=1}^{k} d_{b_i a_{i+1}} - \sum_{i=1}^{k} d_{a_i b_i}. \tag{1}$$

If $\Delta(\boldsymbol{\pi}, \boldsymbol{\pi}^*) < 0$, $\boldsymbol{\pi}^*$ is better (with shorter tour length) than $\boldsymbol{\pi}$.

### 3.3 STATE INITIALIZATION

For state initialization, we choose a simple constructive procedure which starts from an arbitrarily chosen begin city $\pi_1$, and iteratively selects a city $\pi_i, 2 \leq i \leq n$ among the candidate (unvisited) cities (added to the end of the partial tour), until forming a complete tour $\boldsymbol{\pi} = (\pi_1, \pi_2, \ldots, \pi_n)$ which serves as the starting state. More precisely, if there are $m > 1$ candidate cities at the $i$th iteration, each candidate city is chosen with probability $\frac{1}{m}$. Using this method, each possible state is chosen as the starting state with an equal probability.

### 3.4 2-OPT LOCAL SEARCH WITHIN SMALL NEIGHBORHOOD

To maintain the generalization ability of our approach, we avoid to use complex OR techniques, such as the $\alpha$-nearness criterion for selecting candidate edges (Helsgaun, 2000) and the partition and merge method for tackling large TSP instances (Taillard & Helsgaun, 2019), which have proven to be highly effective on the TSP, but heavily depend on expert knowledge.

Instead, we choose a straightforward method to search within a small neighborhood. More precisely, the method examines one by one the promising actions with $k = 2$, and iteratively applies the first-met improving action which leads to a better state, until no improving action with $k = 2$ is found. This method is equivalent to the well-known 2-opt based local search procedure, which is able to efficiently and robustly converge to a local optimal state.

### 3.5 TARGETED SAMPLING OF ENLARGED NEIGHBORHOOD VIA MCTS

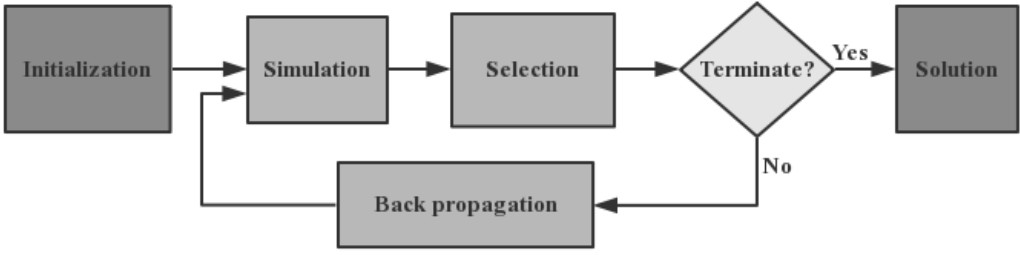

Figure 3: Procedure of the Monte Carlo tree search

Once no improving action is found within the small neighborhood, we close the basic 2-opt local search, and switch to an enlarged neighborhood which consists of the actions with $k > 2$. This method is a generalization of the variable neighborhood search method (Mladenović & Hansen, 1997), which has been successfully applied to many combinatorial optimization problems.

Unfortunately, there are generally a huge number of actions within the enlarged neighborhood (even after eliminating the unpromising ones), being impossible to enumerate them one by one. Therefore, we choose to sample a subset of promising actions (guided by RL) and iteratively select an action to apply, to reach a new state.

Following this idea, we choose the MCTS as our learning framework. Inspired by the works in (Coulom, 2006), (Browne et al., 2012), (Silver et al., 2016) and (Silver et al., 2017), our MCTS procedure (outlined in Fig. 3) consists of four steps, i.e., (1) Initialization, (2) Simulation, (3) Selection, and (4) Back-propagation, which are respectively designed as follows.

**Initialization:** We define two $n \times n$ symmetric matrices, i.e., a weight matrix $\boldsymbol{W}$ whose element $W_{ij}$ (all initialized to 1) controls the probability of choosing city $j$ after city $i$, and an access matrix $\boldsymbol{Q}$ whose element $Q_{ij}$ (all initialized to 0) records the times that edge $(i, j)$ is chosen during simulations. Additionally, a variable $M$ (initialized to 0) is used to record the total number of actions already simulated. Notice that this initialization step should be executed only once at the beginning of the whole process of MDP.

**Simulation:** Given a state $\boldsymbol{\pi}$, we use the simulation process to probabilistically generate a number of actions. As explained in Section 3.2, each action is represented as $\boldsymbol{a} = (a_1, b_1, a_2, b_2, \ldots, a_k, b_k, a_{k+1})$, containing a series of sub-decisions $a_i, 1 \leq i \leq k$ ($k$ is also a variable, and $a_{k+1} = a_1$), while $b_i$ could be determined uniquely once $a_i$ is known. Once $b_i$ is determined, for each edge $(b_i, j), j \neq b_i$, we use the following formula to estimate its potential $Z_{b_i j}$ (the higher the value of $Z_{b_i j}$, the larger the opportunity of edge $(b_i, j)$ to be chosen):

$$Z_{b_i j} = \frac{W_{b_i j}}{\Omega_{b_i}} + \alpha \sqrt{\frac{\ln (M + 1)}{Q_{b_i j} + 1}}. \tag{2}$$

Where $\Omega_{b_i} = \frac{\sum_{j \neq b_i} W_{b_i j}}{\sum_{j \neq b_i} 1}$ denotes the averaged $W_{b_i j}$ value of all the edges relative to city $b_i$. In this formula, the left part $\frac{W_{b_i j}}{\Omega_{b_i}}$ emphasizes the importance of the edges with high $W_{b_i j}$ values (to enhance the intensification feature), while the right part $\sqrt{\frac{\ln (M + 1)}{Q_{b_i j} + 1}}$ prefers the rarely examined edges (to enhance the diversification feature). $\alpha$ is a parameter used to achieve a balance between intensification and diversification, and the term "+1" is used to avoid a minus numerator or a zero denominator.

To make the sub-decisions sequently, we at first choose $a_1$ randomly, and determine $b_1$ subsequently. Recursively, once $a_i$ and $b_i$ are known, $a_{i+1}$ is decided as follows: (1) if closing the loop (connecting $a_1$ to $b_i$) would lead to an improving action, or $i \geq 10$, let $a_{i+1} = a_1$. (2) otherwise, consider the top 10-nearest neighbors of $b_i$ with $Z_{b_i l} \geq 1$ as candidate cities, forming a set $\mathbb{X}$ (excluding $a_1$ and the city already connected to $b_i$). Then, among $\mathbb{X}$ each city $j$ is selected as $a_{i+1}$ with probability $p_j$, which is determined as follows:

$$p_j = \frac{Z_{b_i j}}{\sum_{l \in \mathbb{X}} Z_{b_i l}}. \tag{3}$$

Once $a_{i+1} = a_1$, we close the loop to obtain an action.

Similarly, more actions are generated (forming a sampling pool), until meeting an improving action which leads to a better state, or the number of actions reaches its upper bound (controlled by a parameter $H$).

**Selection:** During above simulation process, if an improving action is met, it is selected and applied to the current state $\boldsymbol{\pi}$, to get a new state $\boldsymbol{\pi}^{new}$. Otherwise, if no such action exists in the sampling

pool, it seems difficult to gain further improvement within the current search area. At this time, the MDP jumps to a random state (using the method described in Section 3.3), which serves as a new starting state.

**Back-propagation:** The value of $M$ as well as the elements of matrices $\boldsymbol{W}$ and $\boldsymbol{Q}$ are updated by back propagation as follows. At first, whenever an action is examined, $M$ is increased by 1. Then, for each edge $(b_i, a_{i+1})$ which appears in an examined action, let $Q_{b_i a_{i+1}}$ increase by 1. Finally, whenever a state $\boldsymbol{\pi}$ is converted to a better state $\boldsymbol{\pi}^{new}$ by applying action $\boldsymbol{a} = (a_1, b_1, a_2, b_2, \ldots, a_k, b_k, a_{k+1})$, for each edge $(b_i, a_{i+1}), 1 \leq i \leq k$, let:

$$W_{b_i a_{i+1}} \leftarrow W_{b_i a_{i+1}} + \beta \left[ \exp \left( \frac{L(\boldsymbol{\pi}) - L(\boldsymbol{\pi}^{new})}{L(\boldsymbol{\pi})} \right) - 1 \right]. \tag{4}$$

Where $\beta$ is a parameter used to control the increasing rate of $W_{b_i a_{i+1}}$. Notice that we update $W_{b_i a_{i+1}}$ only when meeting a better state, since we want to avoid wrong estimations (even in a bad action which leads to a worse state, there may exist some good edges $(b_i, a_{i+1})$). With this back-propagation process, the weight of the good edges would be increased to enhance its opportunity of being selected, thus the sampling process would be more and more targeted.

$\boldsymbol{W}$ and $\boldsymbol{Q}$ are symmetric matrices, thus let $W_{a_{i+1} b_i} = W_{b_i a_{i+1}}$ and $Q_{a_{i+1} b_i} = Q_{b_i a_{i+1}}$ always.

### 3.6 TERMINATION CONDITION

The MCTS iterates through the simulation, selection and back-propagation steps, until no improving action exists among the sampling pool. Then, the MDP jumps to a new state, and launches a new round of 2-opt local search and MCTS again. This process is repeated, until the allowed time (controlled by a parameter $T$) has been elapsed. Then, the best found state is returned as the final solution.

## 4 EXPERIMENTS

To evaluate the performance of our MCTS algorithm, we program it in C language [1], and carry out experiments on a large number of public TSP instances. Notice that the reference algorithms are executed on different platforms (detailed below), being extremely difficult to fairly compare the run times. Therefore, we mainly make comparisons in terms of solution quality (achieved within reasonable runtime), and just list the run-times to roughly evaluate the efficiency of each algorithm.

### 4.1 DATA SETS

Currently there are two data sets widely used by the existing learning based TSP algorithms, i.e., (1) **Set 1** [2], which is divided into three subsets, each containing 10,000 automatically generated 2D-Euclidean TSP instances, respectively with $n = 20, 50, 100$. (2) **Set 2** [3]: which contains 38 instances (with $51 \leq n \leq 318$) extracted from the famous TSPLIB library (Reinelt, 1991). We also use these two data sets as benchmarks to evaluate our MCTS algorithm.

### 4.2 PARAMETERS

As described in Section 3, MCTS relies on four hyper parameters ($\alpha$, $\beta$, $H$ and $T$). We choose $\alpha = 1$, $\beta = 10$, $H = 10n$ ($n$ is the number of cities) as the default settings of the first three parameters. For parameter $T$ used to control the termination condition, we set $T = 75n$ milliseconds for each instance of set 1, and set $T = n$ seconds for each instance of set 2, to ensure that the total time elapsed by MCTS remains reasonable w.r.t. the existing learning based algorithms.

---

[1]Code and results at https://github.com/Spider-scnu/Monte-Carlo-tree-search-for-TSP.
[2]https://drive.google.com/file/d/1-5W-S5e7CKsJ9uY9uVXIyxgbcZZNYBrp/view.
[3]https://wwwproxy.iwr.uni-heidelberg.de/groups/comopt/software/TSPLIB95.

## 4.3 RESULTS ON DATA SET 1

Table 1 presents the results obtained by MCTS on data set 1, with respect to the existing non-learned and learning-based algorithms. In the table, each line corresponds to an algorithm. Respectively, the first nine lines are non-learned algorithms, among which Concorde (Applegate et al., 2006) and Gurobi (Gurobi Optimization, 2015) are two exact solvers, LKH3 (Helsgaun, 2017) is the currently best heuristic, and OR tools are optimization tools released by Google company (Google, 2016). The following six lines are end-to-end ML models (the MCTS algorithm in (Shimomura & Takashima, 2016) did not provide detailed results, thus being omitted), and the final eight lines are hybrid algorithms which use OR method after ML for post-optimization. For the columns, column 1 indicates the methods, while column 2 indicates the type of each algorithm (explained at the bottom of the table), columns 3-5 respectively give the average tour length, average optimality gap in percentage w.r.t. Concorde, and the total clock time used by each algorithm on all the 10,000 instances with $n = 20$. Similarly, columns 6-8, 9-11 respectively give the same information on the 10,000 instances with $n = 50$ and $n = 100$. All the results except ours are directly taken from table 1 of (Joshi et al., 2019) (the order of the algorithms is slightly changed), while unavailable items are marked as "-".

Notice that in the latest learning based algorithms (Kool et al., 2019) and (Joshi et al., 2019), the experiments were carried out either on a single GPU (Nvidia 1080Ti) or 32 instances in parallel on a 32 virtual CPU system (2 × Xeon E5-2630), and the run-time was recorded as the wall clock time used to solve the 10,000 test instances. Similarly, we also run 32 instances in parallel on a 32 virtual CPU system (2 × Intel Xeon Silver 4110 2.1GHz processor, each with eight cores), and report the wall clock time used to solve the 10,000 test instances.

Table 1: Performance of our MCTS algorithm on data set 1, compared to non-learned algorithms (first nine lines), end-to-end ML models (following six lines), and hybrid algorithms which use OR after ML (final eight lines). The optimality gap is computed w.r.t. Concorde (best exact solver).

| Method | Type | TSP20 | | | TSP50 | | | TSP100 | | |
|---|---|---|---|---|---|---|---|---|---|---|
| | | Tour Len. | Opt. Gap. | Time | Tour Len. | Opt. Gap. | Time | Tour Len. | Opt. Gap. | Time |
| Concorde (Applegate et al., 2006) | Exact Solver | 3.84 | 0.00% | 1m | 5.70 | 0.00% | 2m | 7.76 | 0.00% | 3m |
| Gurobi (Gurobi Optimization, 2015) | Exact Solver | 3.84 | 0.00% | 7s | 5.70 | 0.00% | 2m | 7.76 | 0.00% | 17m |
| LKH3 (Helsgaun, 2017) | H | 3.84 | 0.00% | 18s | 5.70 | 0.00% | 5m | 7.76 | 0.00% | 21m |
| Nearest Insertion | H, G | 4.33 | 12.91% | 1s | 6.78 | 19.03% | 2s | 9.46 | 21.82% | 6s |
| Random Insertion | H, G | 4.00 | 4.36% | 0s | 6.13 | 7.65% | 1s | 8.52 | 9.69% | 3s |
| Farthest Insertion | H, G | 3.93 | 2.36% | 1s | 6.01 | 5.53% | 2s | 8.35 | 7.59% | 7s |
| Nearest Neighbor | H, G | 4.50 | 17.23% | 0s | 7.00 | 22.94% | 0s | 9.68 | 24.73% | 0s |
| OR Tools (Google, 2016) | H, S | 3.85 | 0.37% | - | 5.80 | 1.83% | - | 7.99 | 2.90% | - |
| Chr.f. + 2OPT | H, 2OPT | 3.85 | 0.37% | - | 5.79 | 1.65% | - | - | - | - |
| PtrNet (Vinyals et al., 2015) | SL, G | 3.88 | 1.15% | - | 7.66 | 34.48% | - | - | - | - |
| PtrNet (Bello et al., 2017) | RL, G | 3.89 | 1.42% | - | 5.95 | 4.46% | - | 8.30 | 6.90% | - |
| S2V-DQN (Khalil et al., 2017) | RL, G | 3.89 | 1.42% | - | 5.99 | 5.16% | - | 8.31 | 7.03% | - |
| GAT (Deudon et al., 2018) | RL, G | 3.86 | 0.66% | 2m | 5.92 | 3.98% | 5m | 8.42 | 8.41% | 8m |
| GAT (Kool et al., 2019) | RL, G | 3.85 | 0.34% | 0s | 5.80 | 1.76% | 2s | 8.12 | 4.53% | 6s |
| GCN (Joshi et al., 2019) | SL, G | 3.86 | 0.60% | 6s | 5.87 | 3.10% | 55s | 8.41 | 8.38% | 6m |
| GNN (Nowak et al., 2017) | SL, BS | 3.93 | 2.46% | - | - | - | - | - | - | - |
| PtrNet (Bello et al., 2017) | RL, S | - | - | - | 5.75 | 0.95% | - | 8.00 | 3.03% | - |
| GAT (Deudon et al., 2018) | RL, S | 3.84 | 0.11% | 5m | 5.77 | 1.28% | 17m | 8.75 | 12.70% | 56m |
| GAT (Deudon et al., 2018) | RL, G, 2OPT | 3.85 | 0.42% | 4m | 5.85 | 2.77% | 26m | 8.17 | 5.21% | 3h |
| GAT (Deudon et al., 2018) | RL, S, 2OPT | 3.84 | 0.09% | 6m | 5.75 | 1.00% | 32m | 8.12 | 4.64% | 5h |
| GAT (Kool et al., 2019) | RL, S | 3.84 | 0.08% | 5m | 5.73 | 0.52% | 24m | 7.94 | 2.26% | 1h |
| GCN (Joshi et al., 2019) | SL, BS | 3.84 | 0.10% | 20s | 5.71 | 0.26% | 2m | 7.92 | 2.11% | 10m |
| GCN (Joshi et al., 2019) | SL, BS* | 3.84 | 0.01% | 12m | 5.70 | 0.01% | 18m | 7.87 | 1.39% | 40m |
| MCTS (ours) | RL, 2OPT | **3.8303** | **-0.0075%** | 8m | **5.6906** | **-0.0212%** | 20m | **7.7631** | **-0.0178%** | 40m |

**H**: Heuristic, **SL**: Supervised Learning, **RL**: Reinforcement Learning, **G**: Greedy, **S**: Sampling,
**BS**: Beam search, **BS***: BS and shortest tour heuristic, **2OPT**: 2-opt local search.

As shown in Table 1, the exact solvers and LKH3 obtain good results on all the test instances, while the remaining six non-learned algorithms perform overall poorly. Among the learning based algorithms, in terms of solution quality the hybrid algorithms which combines ML and OR perform clearly much better than the end-to-end ML models, although much more computational times are required. Finally, compared to these existing methods, our MCTS algorithm performs quite well, which succeeds in matching or improving the best known solutions (reported by Concorde) on most

of these instances, corresponding to an average gap of **-0.0075%**, **-0.0212%**, **-0.0178%** [4] respectively on the three subsets with $n = 20, 50, 100$. The computation times elapsed by MCTS remains reasonable (40 minutes for the 10,000 instances with $n = 100$), close to the latest (currently best) learning based algorithm (Joshi et al., 2019).

### 4.4 RESULTS ON DATA SET 2

This set contains 38 instances ($51 \leq n \leq 318$) extracted from the TSPLIB. For each of these instances, we list in Table 2 the optimal result (reported by Concorde within 1 hour for each instance), the results reported by S2V-DQN (Khalil et al., 2017) and our MCTS algorithm (indicated in **bold** if reaching optimality). Notice that S2V-DQN did not clearly report the computational time, while $n$ seconds is allowed by our MCTS to solve each instance with $n$ cities (the 38 instances are executed sequentially, occupying a core of an Intel Xeon Silver 4110 2.1GHz processor).

As shown in the table, S2V-DQN reaches optimality on only one instance (berlin52), corresponding to a large average gap (w.r.t. the optimal solutions) of $4.75\%$. For comparison, our MCTS succeeds in matching the optimal solutions on 28 instances, corresponding to a much smaller average gap ($0.21\%$). Furthermore, MCTS dominates S2V-DQN on all these instances only except two instances pr226 (worse than S2V-DQN) and berlin52 (with equal results), clearly demonstrating its superiority over S2V-DQN.

Table 2: Performance of MCTS on data set 2 compared to S2V-DQN (Khalil et al., 2017).

| Instance | OPT | S2V-DQN | MCTS | Instance | OPT | S2V-DQN | MCTS |
|---|---|---|---|---|---|---|---|
| eil51 | 426 | 439 | **426** | berlin52 | 7542 | **7542** | **7542** |
| st70 | 675 | 696 | **675** | eil76 | 538 | 564 | **538** |
| pr76 | 108159 | 108446 | **108159** | rat99 | 1211 | 1280 | **1211** |
| kroA100 | 21282 | 21897 | **21282** | kroB100 | 22141 | 22692 | **22141** |
| kroC100 | 20749 | 21074 | **20749** | kroD100 | 21294 | 22102 | **21294** |
| kroE100 | 22068 | 22913 | **22068** | rd100 | 7910 | 8159 | **7910** |
| eil101 | 629 | 659 | **629** | lin105 | 14379 | 15023 | **14379** |
| pr107 | 44303 | 45113 | **44303** | pr124 | 59030 | 61623 | **59030** |
| bier127 | 118282 | 121576 | **118282** | ch130 | 6110 | 6270 | **6110** |
| pr136 | 96772 | 99474 | **96772** | pr144 | 58537 | 59436 | 58763 |
| ch150 | 6528 | 6985 | **6528** | kroA150 | 26524 | 27888 | **26524** |
| kroB150 | 26130 | 27209 | **26130** | pr152 | 73682 | 75283 | **73682** |
| u159 | 42080 | 45433 | **42080** | rat195 | 2323 | 2581 | 2324 |
| d198 | 15780 | 16453 | 15785 | kroA200 | 29368 | 30965 | **29368** |
| kroB200 | 29437 | 31692 | 29438 | ts225 | 126643 | 136302 | **126643** |
| tsp225 | 3916 | 4154 | 3926 | pr226 | 80369 | 81873 | 83467 |
| gil262 | 2378 | 2537 | 2391 | pr264 | 49135 | 52364 | **49135** |
| a280 | 2579 | 2867 | **2579** | pr299 | 48191 | 51895 | 48195 |
| lin318 | 42029 | 45375 | 42314 | linhp318 | 41345 | 45444 | 42305 |

The average gap of S2V-DQN vs. OPT is $4.75\%$, while the average gap of MCTS vs. OPT is **0.21%**.

Overall, we think the experimental results on above two data sets clearly show the potential of our MCTS on the TSP.

## 5 CONCLUSION

This paper newly develops a novel flexible paradigm for solving TSP, which combines OR and ML in a variable neighborhood search strategy, and achieves highly competitive performance with respect to the existing learning based TSP algorithms. However, how to combine ML and OR reasonably is still an open question, which deserves continuous investigations. In the future, we would try more new paradigms to better answer this question, and extend the work to other combinatorial optimization problems.

---

[4]On many instances, the best known solutions reported by Concorde are not strictly optimal (confirmed in (Joshi et al., 2019), possibly due to round-off reasons), which could be slightly improved ($< 10^{-2}$) by our MCTS algorithm. On the 10,000 instances with $n = 20, 50, 100$, MCTS respectively improves (with different permutations of the cities) 1094, 3682, 6404 and misses 0, 0, 1273 best known solutions, while matching the best known solutions on all the remaining instances.

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
