# OpenReview forum: "Targeted sampling of enlarged neighborhood via Monte Carlo tree search for TSP"
_ICLR.cc/2020/Conference — Reject_

### Official Review · AnonReviewer3 · 2019-10-08
**Official Blind Review #3**

**Rating:** 1

**Review:**

This paper proposes a new RL-based algorithm for solving the traveling salesman problem (TSP). Its main component is the combination of OR-based 2-opt search and learning-based k-opt search. Monte Carlo tress search is employed to train the learning-based k-opt search. The experimental result suggests state-of-the-art performance over existing RL-based solvers.

My main criticism is about the positioning of the paper in the literature. This algorithm is more of an "online" RL algorithm with learning done for each TSP instance separately, while the compared "offline" RL algorithms perform learning over the TSP algorithm (and training dataset). Regarding this matter, I failed to understand the advantage or promise of the proposed algorithm over the existing solvers such as Concorde. The motivation in the paper states that existing OR algorithms are "very complicated and consist of many hand-crafted rules and heavily rely on expert knowledge, thus being difficult to generalize to other combinatorial optimization problems", but this algorithm seems to suffer the same problems. In particular, one could even assert that the proposed algorithm is an instance of the Tabu search method [1], which is based on keeping a record of actions taken and penalizing to revisit it.

Furthermore, I think the experiment lacks some detail regarding the complexity of the algorithm. The comparison of the proposed algorithm to the existing RL-based algorithms is slightly unfair, given that the proposed algorithm is implemented in C and others are implemented in Python (except S2V-DQN). For instance, when MCTS and the method of (Joshi et al., 2019) both take 40 minutes on the TSP-100 dataset, it is likely that MCTS is slower when implemented in Python. I would suggest the authors to avoid misleading the readers by including this fact in the paper. I also suggest providing the number of iterations it took to achieve the reported results since one could compare it to other local search methods. Furthermore, the experiments seem to lack the details on the number and type of GPUs used for obtaining the reported results.

[1] Zachariasen and Dam, Tabu Search on the Geometric Traveling Salesman Problem, Meta-Heuristics. Springer 1966


**Experience Assessment:**

I have read many papers in this area.

**Review Assessment: Checking Correctness Of Derivations And Theory:**

N/A

**Review Assessment: Checking Correctness Of Experiments:**

I assessed the sensibility of the experiments.

**Review Assessment: Thoroughness In Paper Reading:**

I made a quick assessment of this paper.

---

### Official Review · AnonReviewer1 · 2019-10-22
**Official Blind Review #1**

**Rating:** 1

**Review:**

The paper presents a machine-learning based heuristic for solving traveling
salesman problems. In particular, MCTS is used to explore a large neighbourhood.
The authors present their approach and evaluate it empirically.

The presented approach is interesting; a few details could be described in more
detail and motivated better (for example how the particular functional form for
estimating the potential Z of an edge was chosen). but in general the paper is
well-written.

The main part where the paper falls short is the experimental evaluation. The
authors state that the reference algorithms were executed on different
platforms, even though at least some of them are publicly available and the
authors could have run them themselves. In their own experimental setup, the
authors overload the machines by solving instances on each hyper-threaded
logical core instead of the physical cores for no apparent reason. Running on
logical cores like this leads to significantly longer runtimes. This
experimental setup is changed for instance set 2 for no apparent reason.

The authors claim to improve on the optimal solution that concorde finds,
confirmed by a non-peer-reviewed paper, without providing a justification -- if
this is due to rounding errors, are the found tours the same and just the length
computation is flawed? Or are the tours different?

Tables 1 and 2 present results in completely different formats. This makes it
unnecessarily hard to compare results. In particular, Table 2 presents no run
times.

Finally, the instances used to evaluate the approach seem relatively easy.
TSPlib contains many more instances that are more challenging to solve, with
hundreds to thousands of cities. Even on the relatively small instances, the
presented approach is often an order of magnitude slower than the exact solver
concorde -- why would I want to use the presented approach in a practical
setting?

In summary, I feel that the paper cannot be accepted in its current form.


**Experience Assessment:**

I have published one or two papers in this area.

**Review Assessment: Checking Correctness Of Derivations And Theory:**

I assessed the sensibility of the derivations and theory.

**Review Assessment: Checking Correctness Of Experiments:**

I assessed the sensibility of the experiments.

**Review Assessment: Thoroughness In Paper Reading:**

I read the paper at least twice and used my best judgement in assessing the paper.

---

### Official Review · AnonReviewer2 · 2019-10-23
**Official Blind Review #2**

**Rating:** 3

**Review:**

The paper proposes an algorithm for the Travelling Salesman Problem that starts with a random tour and iteratively improves it using 2-opt local search, followed by Monte Carlo Tree Search with k-opt moves to search a larger neighborhood of solutions. Results show that the algorithm matches the optimal objective value on a dataset of 2D Euclidean TSP instances with 20, 50, and 100 cities, and achieves better objective value than S2V DQN on a subset of instances from TSPLIB with 50 to 318 cities. It is interesting that the algorithm is fairly simple (apart from the definition of the local moves) and still able to achieve competitive objective values on small instances.

The rating is a weak reject for the following reasons:
1. Domain knowledge: The proposed algorithm has significant domain knowledge about TSP hand-coded into it via the choice of 2-opt local search as well as the k-opt moves with top-10 nearest neighbor filtering used in MCTS. These types of moves are same/similar to the expert-designed ones used in state-of-the-art local search algorithms like LKH. This can give the proposed algorithm a significant advantage over S2V-DQN which does not have such built-in knowledge. So the comparison to S2V-DQN is unfair and improved results may not be due to any differences in the learning approaches at all. Perhaps a more generic definition of local moves that is not so TSP-specific would be a fairer comparison. This issue is particularly important because the definition of “local moves” appropriate for a problem domain is one of the main challenges in applying local search successfully. Bypassing that challenge can simplify the problem.

2. Baselines: Better comparisons to baselines are needed in Table 2. S2V-DQN was published two years ago, and since then many other learning-based have been proposed (which are referenced in the paper), but they are not included in Table 2. It would also be good to discuss non-learning local search algorithms that use same/similar local moves as the proposed algorithm and how they perform.

3. Scalability to large instances: TSP experts consider problems with hundreds of cities as practically “solved” by the current state-of-the-art solvers like Concorde and LKH, and it is not at all clear what benefit learning can provide on such small instances. Much larger instances (e.g., 10^4-10^6 cities) would have to be considered for improvements over the state-of-the-art to become plausible with learning. So if the goal is to improve on the state-of-the-art with learning (even if it is a long-term goal), then it is important to consider whether the proposed approach has a plausible path for scaling up to such sizes. But like most papers on learning-for-TSP, this paper does not consider scalability.

Misc comments:
It would be useful to include the exact running times of Concorde in Table 2. The results given are “reported by Concorde within 1 hour for each instance”, but it would be good to know what the actual running time is.
Typo: “sub-decisions sequently”


**Experience Assessment:**

I have published one or two papers in this area.

**Review Assessment: Checking Correctness Of Derivations And Theory:**

I did not assess the derivations or theory.

**Review Assessment: Checking Correctness Of Experiments:**

I assessed the sensibility of the experiments.

**Review Assessment: Thoroughness In Paper Reading:**

I read the paper thoroughly.

---

### Decision · Program_Chairs · 2019-12-19

**Decision:**

Reject

**Comment:**

This paper contributes to the recently emerging literature about applying reinforcement learning methods to combinatorial optimization problems.
The authors consider TSPs and propose a search method that interleaves greedy local search with Monte Carlo Tree Search (MCTS).
This approach does not contain learned function approximation for transferring knowledge across problem instances, which is usually considered the main motivation for applying RL to comb opt problems.

The reviewers state that, although the approach is a relatively straight-forward combination of two existing methods, it is in principle somewhat interesting.
However, the experiments indicate a large gap to SOTA solvers for TSPs.
No rebuttal was submitted.

In absence of both SOTA results and methodological novelty, as assessed by the reviewers and my owm reading, I recommend to reject the paper in its current form.